# Raising the Resilience of Industrial Manufacturers through Implementing Natural Gas-Fired Distributed Energy Resource Systems with Demand Response

**Anatolyy Dzyuba \*, Irina Solovyeva \***  **and Aleksandr Semikolenov**

Department of Economics and Finance of the Higher School of Economics and Management,
South Ural State University, Chelyabinsk 454091, Russia; semikolenov83@yandex.ru
**\*** Correspondence: dziubaap@susu.ru (A.D.); solovevaia@susu.ru (I.S.); Tel.: +7-922-636-59-78 (A.D.)

**Abstract:** The use of relatively small-scale distributed electric power generation sources is one of the key focus areas in the development of global industry and regional power generation. By integrating distributed generation sources into their on-site energy infrastructure, industrial consumers gain new characteristics and possibilities as entities of the power system that do not only consume power, but in fact can flexibly generate and deliver electricity to local and even centralized grids. This type of entity is called a distributed energy resource system with demand response (Russian: 'active energy complex'). The purpose of this study is to lay the methodological foundation for the use of distributed energy resource systems with demand response in industrial sites under existing gas and power market conditions and for ensuring the synchronization of parameters that is necessary for managing complex energy consumption. This article provides an empirical study of the principles of the natural gas pricing under the demand volatility of regional markets and the Russian Mercantile Exchange. The article outlines the key drivers, as identified by the authors, that impact gas consumption by a distributed energy resource system, including demand characteristics, limitations and capacity of the gas network and the mode of gas consumption by an industrial enterprise and its generator. Accounting for all of these factors is essential for effective management and proper operational adjustment of a distributed energy resource system with demand response. The result of the study is a proprietary model and a tool for the management of distributed energy resource systems in integration with the gas demand management, which analyze the internal and external parameters of the industrial entity's operations and its distributed energy resource system, as well as factors existing in the integrated distributed energy system where the consumer is able to buy natural gas in various market segments. The proprietary tool of distributed energy resource system management is based on the centralized control system, which combines performance analytics, operational scheduling of production and the distributed energy resource system, price planning for the wholesale and retail power markets, regional gas markets and exchange, monitoring all elements of the system, and assessment of different active energy management scenarios under various external and internal conditions impacting production and energy demand. Our proprietary tool has been successfully tested in a typical industrial site and was reported to deliver a significant electricity and gas cost-saving effect, which amounted to an 18 percent reduction in the total energy costs of the company, or more than USD 2.6 million per year. The resulting saving effect can recoup the costs of investing in a distributed energy resource system, including construction and installation of the local grid and automation infrastructure, and can be obtained in any country of the world.

**Keywords:** industrial energy; natural gas consumption; gas demand management; gas industry; distributed generation systems; gas demand schedules; energy efficiency; price-dependent consumption

## 1. Introduction

Over the last century, the process of global economic development has been continuously accompanied by the growth in the consumption of fuel and energy resources. Global

industrialization, which began in different countries between the late 19th and early 20th centuries, is primarily characterized by the rising, higher-intensity consumption of energy resources due to burgeoning manufacturing, extensive use of steam engines, expansion of railways, large-scale electrification of industry and agriculture, labor mechanization, and the growth of energy-intensive production industries [1]. Under these conditions, the availability of energy resources as a means of sustainable energy supply has been inextricably involved in making decisions about establishing and developing any enterprises or even communities. Communities with high energy supply gradually grew into large economic and industrial hubs establishing a platform for further economic, technological, infrastructural, administrative, and scientific development. On the contrary, those entities which lacked sustainable fuel and energy potential lagged significantly in terms of their economic and industrial development [2].

Figure 1 shows the mix of the global energy balance in 1900–2020. Over the 120 years under study, the global energy balance mix changed several times due to various technological and economic factors. For instance, the technology factor impacted the development of extraction and processing technologies of some energy resources. An example of a technological shift is the development of coal mining technology and its transportation over long distances by rail—these have significantly expanded coal application across industries. Examples of economic factors include an increase in oil prices and a decrease in prices for renewable energy, which underlies the energy transition concept. In 2020, more than 3.8% of consumed energy was renewable, whereas in 2006 this was less than 0.5% [3]. Implementation of an energy transition policy in the near future will also impact on the global energy balance mix, changing the concept of energy consumption in most countries around the world.

The consumption of energy resources and electrical energy in different countries of the world varies not only in terms of scale, but also efficiency. The efficiency of fuel and energy consumption is reflective of how well the energy resources are utilized in the production of certain goods and/or services. For example, manufacturers in Russia and Finland consume energy resources differently, because in Russia, efficiency is impaired by less efficient manufacturing equipment, obsolete grids and lack of consumption and quality monitoring. On the contrary, in Finland, energy resources are utilized by high-quality and high-efficiency equipment with proper energy accounting and consumption control. Therefore, Finland is able to produce more than Russia using the same amount of energy resources.

The comparative level of energy efficiency of different economies is traditionally measured as the energy intensity of GDP [4,5], which reflects energy consumption by a country for the production of one dollar in GDP (PPP). Figure 1 shows a map of the energy intensity of GDP (PPP) in different countries of the world in 2020. As follows from the map, different economies demonstrate significant variation in terms of their energy intensity. For example, Germany, Denmark and the Netherlands spend 2.5 times less energy per dollar of GDP compared to Russia, Uzbekistan or Zimbabwe [6,7]. This emphasizes the lag in some countries in terms of consumption efficiency as well as the urgency to look for opportunities to improve their energy efficiency [8]. In such conditions, one of the key areas for improving the corporate sustainability of the industrial business in Russia is the development of innovative mechanisms in the field of increasing the level of energy efficiency of the economy, which, among other things, are implemented on the basis of business models of use of natural resources, gas, with consumption.

Figure 2 shows the evolution of energy-saving and energy efficiency technologies. Energy efficiency has been given more focus along with the increasing efficiency of energy-consuming industrial equipment, followed by the integration of the energy efficiency agenda into large-scale initiatives undertaken on the national level. With the development of digital technologies, energy efficiency has incorporated the digital aspect in such concepts as the smart grid, smart meters, etc. The global energy transition shift and the quest to bring down the price of the renewable energy have accelerated the application of energy saving technology, renewable energy sources and distributed energy systems.

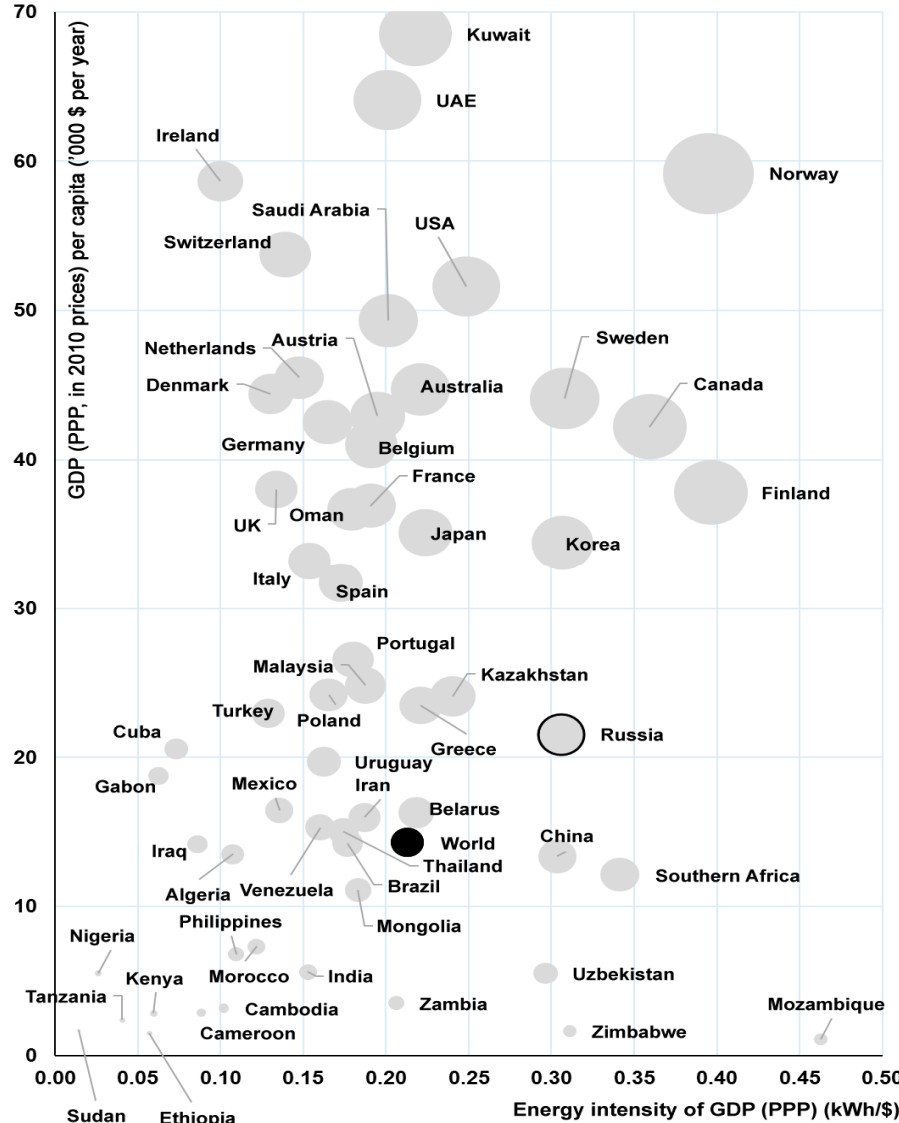

**Figure 1.** Map of energy intensity of GDP in 2020 (PPP) (circle size corresponds to energy consumption per capita) [8,9].

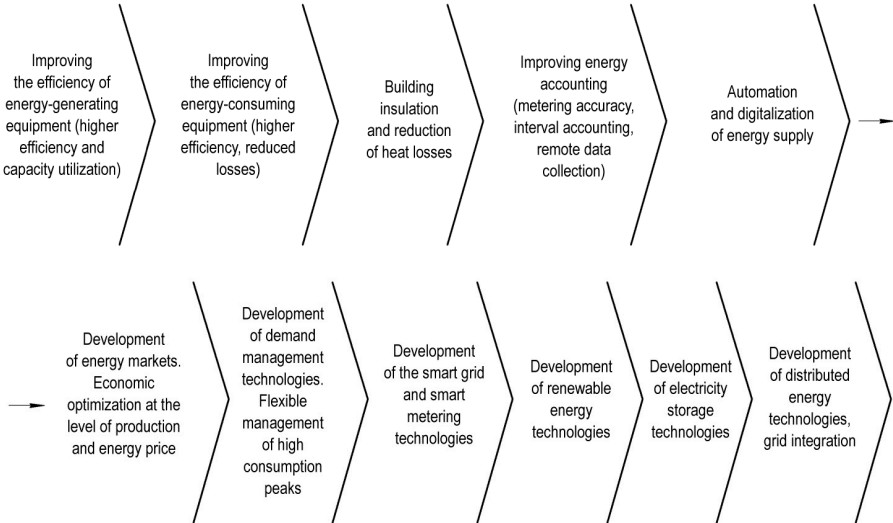

**Figure 2.** The evolution of energy saving and energy efficiency technologies.

According to the International Renewable Energy Agency (IRENA), new distributed generation capacity is expected to exceed new commissions of centralized generation by 86% already by 2023 (100 GW of conventional vs. 86 GW of distributed generation), reaching more than 209% by 2026 and 330% by 2030 [8]. These optimistic estimations allow one to assume that distributed generation will be one of the core levers of energy efficiency in industry because distributed energy resource (DER) systems are installed near large, energy-intensive facilities, primarily at industrial enterprises, and serve to reduce the costs of the manufacturers not only in terms of power generation, but also in terms of transmission and ultimate consumption [10].

Figure 3 shows the key focus areas of energy efficiency and energy reliability in the global energy system in the context of modern technological trends.

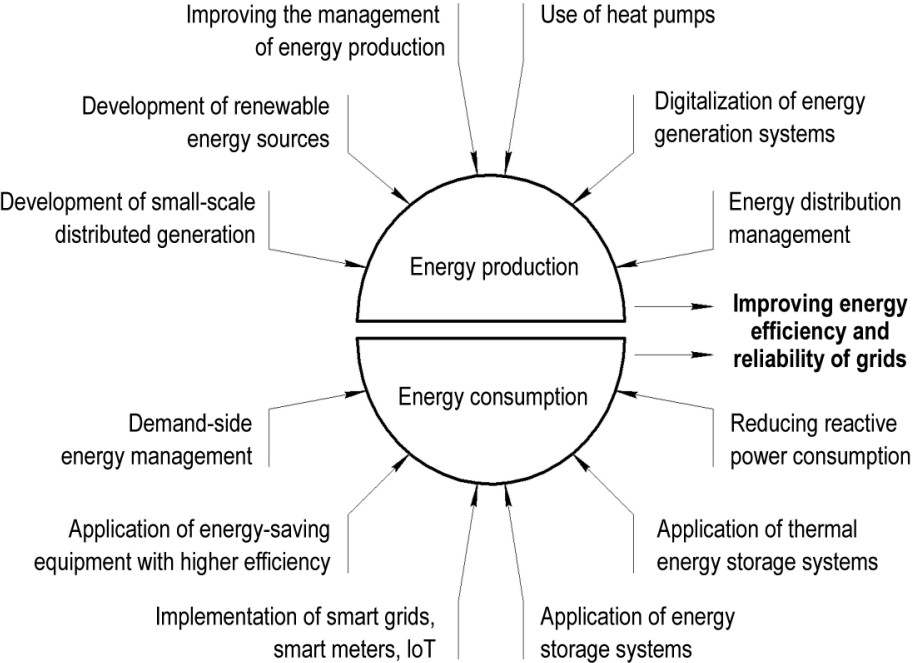

**Figure 3.** Key focus areas of energy efficiency and energy reliability in the global energy system in the context of modern technological trends.

The potential for increasing energy efficiency does not lie on the demand side alone, but applies to generation too. In our opinion, one of the key areas of the energy efficiency focus in industry should be small-scale distributed energy systems integrated with advanced demand-side management (e.g., managing energy and gas demand). Installed at industrial sites, these DER systems will be integrated into the centralized energy and gas grids later in future.

Russia's unique fuel and energy potential defines the specifics of domestic energy consumption. Figure 4 shows the mix of electricity generation from fossil fuels in select countries in 2020. As seen from the figure below, the share of natural gas in the total fossil fuels used in Russia in 2020 was 75.5%, whereas in other countries this indicator was much lower, with the global average being 33.4%. When analyzing the data in Figure 4, it should be taken into account that fossil fuel generation is decreasing in many countries along with the expansion of renewables. Given the scale of electricity consumption in Russia, the scale of natural gas used to generate electricity is also significant.

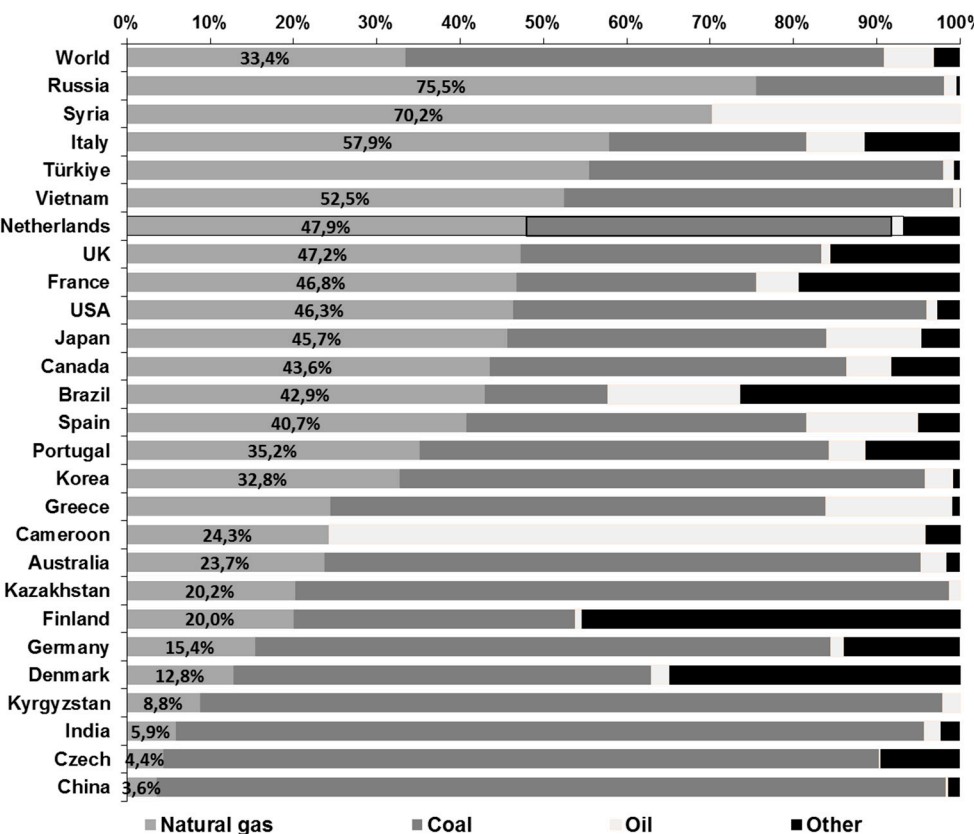

**Figure 4.** The mix of electricity generation from fossil fuels in select countries in 2020 [11,12].

Russia's domestic production of natural gas defines the gas price in the domestic market. Figure 5 below shows the price of natural gas in various countries of the world in 2020. Russia has one the cheapest gas prices in the world, which is more than 10× lower than in Sweden, Spain, The Netherlands, Italy, France, etc., and more than 5× lower than in most other countries.

The price of electricity in Russia's domestic market is largely driven by the domestic price of natural gas and its high share in the energy generation mix [13]. Russia's domestic price of electricity is practically the lowest in the world. The average price of electricity in Russia is USD 0.08 per 1 kilowatt-hour, which is more than 3× less than in Germany, UK, Italy, Denmark, etc. The low cost of electricity supplied to all sectors of the Russian economy coupled with the low cost of natural gas presents a serious impediment to introducing energy-saving and energy efficiency solutions and renewables [11].

Figure 6 shows a diagram of the shares of electricity generation from renewable energy sources in 2020 by country. As follows from the diagram, the share of renewables in the energy generation mix exceeded 40% in the UK and Germany in 2020, with a smaller share in Belgium and The Netherlands (25%) and an almost insignificant 0.32% in Russia.

As distributed energy systems are becoming more common, the low price of electricity in many countries is likely to urge manufacturers to implement DER systems based on natural gas. That being said, it is therefore of practical interest to review the application of fossil-fuel-based DER systems. Given the high share of natural gas consumption in the total electricity generation mix of the distributed energy systems operated in Russia, such a study should be based on Russian cases from the fuel and energy industry. In addition, the application of DER should take into account the fact that it will be operated in the context of Russia's existing power and gas market environment. The presented case models of generation and demand management can also be applied to the other markets because the Russian electricity market model was developed based on Nord Pool, a pan-European power exchange.

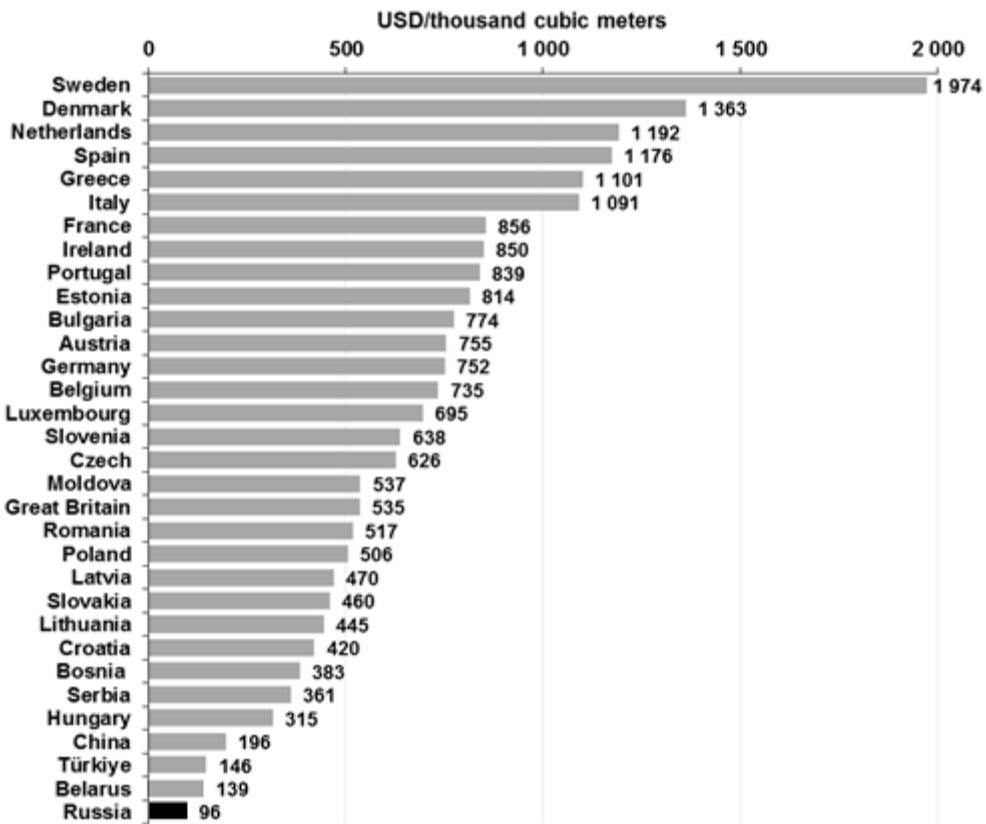

**Figure 5.** The price of natural gas in different countries of the world in 2020 (in USD per 1000 cubic meters) [14].

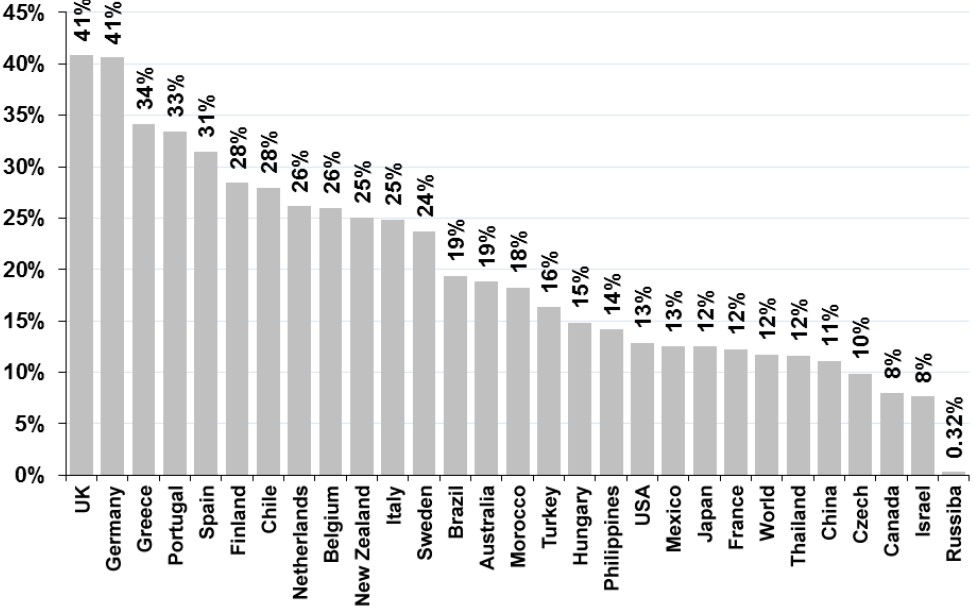

**Figure 6.** The share of electricity generation from renewable energy sources in 2020 by country [7].

The current international research in the field of distributed generation is largely focused on modeling the operation of distributed energy systems with centralized grids and on improving sustainability of the power supply. Examples include the studies carried out by Matos, S.P.S. [15] and Beltran, J.C. [16], which focus on increasing the stability of the parallel operation of distributed generation systems and centralized grids. Research by Sandhya, K. [17] and Martinez, S.D. [18] are directed at ways to reduce failures and

improve the quality of power supply for distributed generation systems. The studies by Anuradha, K.B.J. [19] and Baghbanzadeh, D. [20] point to the importance of loss reduction in the operation and distribution of electricity generated by distributed energy systems.

## 2. Literature Review of the Topic of Natural Gas-Fired Distributed Energy Resource (DER) Systems with Demand Response

Much modern international research in the field of application of distributed generation systems focuses on integrating digital control and management tools into distributed energy systems. These include studies by Li [21], Kakran [22] and Howlader [23] on smart management of distributed generation systems. The work of Belmahdi [24] and Valencia [25] are focused on digital modeling of the operating modes of distributed generation systems. Menke [26] and Rahiminejad [27] study the possibilities of digital monitoring and optimal scheduling for distributed generation systems.

Given the high degree of development of electricity markets in many countries, a significant part of international research in the field of distributed generation is devoted to the integration of distributed energy systems with the power market pricing mechanisms. Examples include works by Abdulkareem [28] and Craig [29]. Studies by Yu [30] and Liu [31] outline the business modeling of the distributed generation systems in the studied energy markets. The works of Lin [32] and Kumar [33] model the operation of distributed generation systems, taking into account the signals and specifics of the respective electricity markets.

In terms of their operating structure, renewable energy sources pertain to distributed generation technologies. The intense development of renewables has sprung active research on their application in distributed generation. Examples of such works include the research by Zhang [34], Samper [35] and Garlet [36] focusing on distributed photovoltaic generation. DER-based distributed energy systems intertwine with the use of industrial energy storage systems, which is yet another focus area of the current research. The research works devoted to the study of energy storage and distributed generation include the studies by Ahmadi [37] and Yanine [38]. Another area of the international research is directed to distributed generation based on renewable energy sources in the areas with the centralized grid capacity limitations (e.g., works by Das [39] and Thopil [40]. Another focus area of the international researchers is optimization of the operating modes of distributed generation systems as distributed sources of energy (e.g., works by Abdmouleh [41] and Monteiro [42]).

The digitalization of the energy industry has led to the development of demand response technologies and their adoption in many energy systems worldwide, which has become another area of scientific research and interest in the field of distributed generation. Among the works focusing on the management of electrical loads of distributed generation systems and its link to the demand response are the studies by Wang [43], Howlader [23] and Nakada [44]. A number of studies have been conducted on the issue of grid capacity enhancement through the adoption of distributed generation systems and demand response (Nejad, H.C. [45] and Poudineh, R. [46]). Another significant part of the research in the field of distributed energy is directed to the digitalization of the processes of managing the demand for electricity by consumers that use distributed generation sources. These include the works of Wang [43], Viana [47] and Jiang [48].

It should be noted that the geography of the global research in the field of distributed energy is very extensive, which means that the number of scientific papers published on this topic is significant. Originally beginning in North America and Europe [49–51], the global research on distributed generation has now expanded to Asia, Africa and the Middle East [52–54], which demonstrates its important prospects for the theoretical and practical development of distributed generation across the world.

Considering that the bulk of distributed generation systems in use are fired by natural gas, these units generate electricity in the cogeneration mode (i.e., they simultaneously generate electrical and thermal energy). Russian researchers Dormidonov [55], Safonov [56] and Pivnyuk [57] estimate the efficiency of cogeneration of electricity in distributed energy systems and pinpoint their advantages compared to the electricity-only generation sources.

The works by Makarova [58], Lachkov [59] and Nalbandyan [60] study the efficiency of distributed energy sources based on cogeneration. In general, the researchers emphasize the advantages of using energy cogeneration systems, primarily due to their increased efficiency and covering the demand not only for electrical, but also for thermal energy.

Taking into account the novelty of distributed generation for the Russian power industry, a number of Russian researchers have directed their works to describing the novelty and development prospects of the country's energy system with the advent of distributed generation. These include the studies by Smirnov [61], Polomoshin [62], Volkov [63], etc. Some researchers focus on the problems of introducing distributed generation into Russia's centralized energy system, which is associated with the complex management of the Russian Grid, the inconsistent nature of the distributed generation systems and the possible increase in the incident rate due to the integration of distributed generation sources.

The concept of 'distributed energy resource (DER) systems' was first introduced in Russia in the Government Act 320 'On the Amendments to the Government Acts Concerning Distributed Energy Resource Systems' issued on 21/3/2020 [64], starting a new area of Russian research in this domain. Examples of the works focusing on DER systems in the context of the Russian grid include the studies of Voropay [65,66], Protsenko [67], Datsko [67] and Byk [68] who study the implications of adopting DER systems within the Russian grid. The studies carried out by Anikeeva [69], Semerenko [70] and Karanina [71] outline different aspects of introducing DER systems into the operations of the Russian electricity distribution network.

The review of the current international and Russian research in the field of application of distributed energy systems in industrial enterprises allows us to conclude that the currently held approaches and assumptions do not take into account the following important factors:

(1) There is no methodological approach to the complex management of costs for electricity and natural gas in DER systems implemented by industrial and commercial facilities in Russia;

(2) The uneven nature of the natural gas demand curves is not properly accounted for in the operation of DER systems;

(3) Changes in the complex demand for natural gas and electricity are not properly accounted for in scheduling DER systems;

(4) The pricing in the wholesale and retail electricity markets is not properly accounted for in the operation of DER systems;

(5) No methods developed for analyzing the price parameters of the electricity purchased by industrial and commercial entities that have direct connection to the networks of electricity producers; and

(6) No algorithm for the integrated management of behind-the-meter DER systems in the context of the existing power (capacity) and natural gas markets.

## 3. The Specifics of Industries' Demand Curves for Natural Gas

The gas demand curves of industrial enterprises and distributed generation systems as well as the electricity demand curves of industrial enterprises are very unbalanced. Figure 7 shows examples of daily consumption of natural gas by industrial enterprises from various sectors. The diagrams below suggest that the consumption of natural gas is very seasonal and volatile in the context of shorter periods, in particular, within a month or a week. Manufacturers from different sectors have different degrees of gas demand fluctuations, which are related to the specifics of their gas-powered equipment in their sector as well as other specific factors that impact their demand for natural gas.

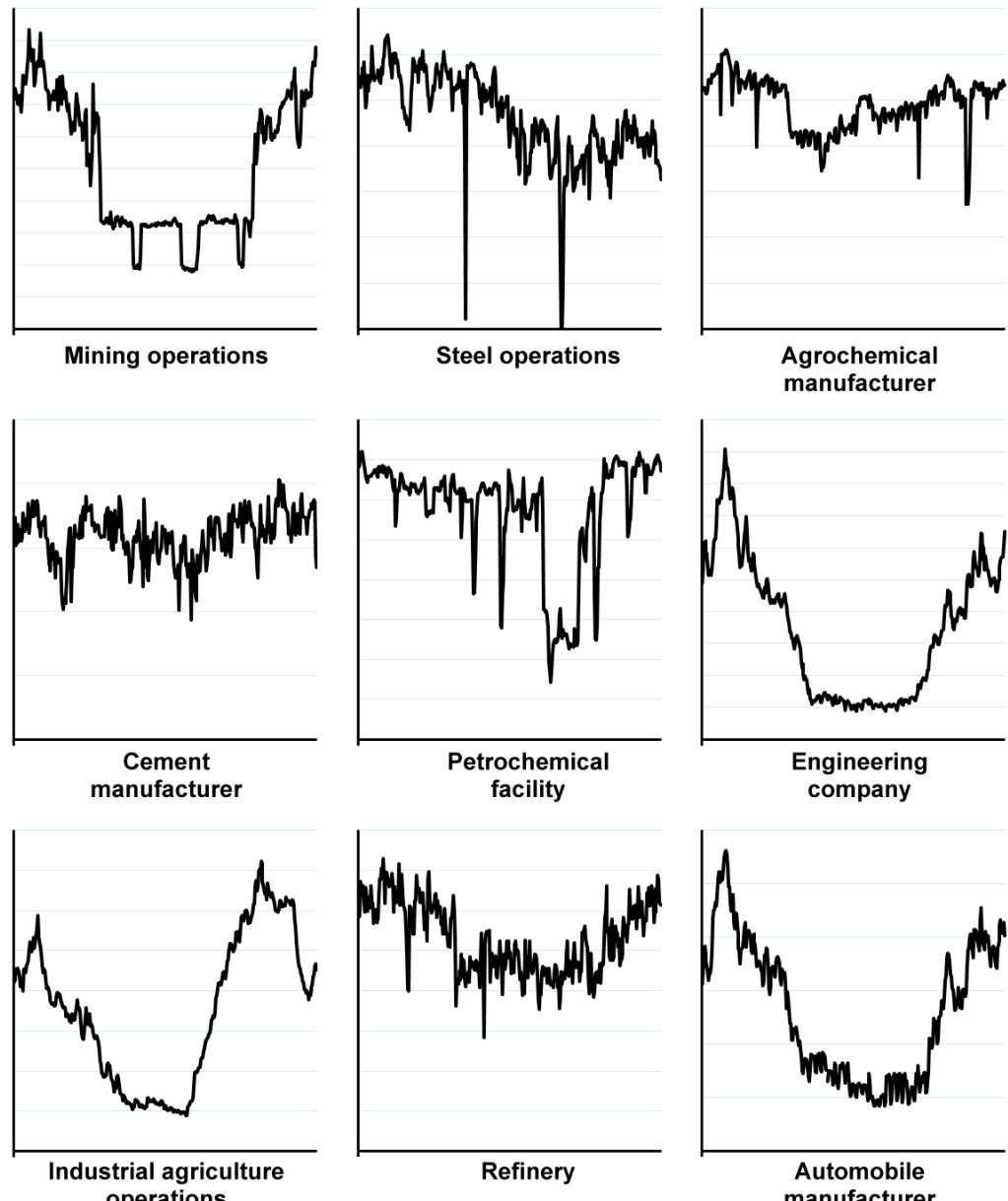

**Figure 7.** Daily natural gas demand curves of industrial enterprises from various sectors.

The gas demand curves of industrial enterprises are similar to their electricity demand curves, which is due to the following factors:

- Natural gas as well as electricity are supplied from the centralized grids whenever the demand appears;
- Natural gas, like electricity, cannot be stored on a large scale at an industrial enterprise in order to balance its demand from the grid in the context of continuous gas consumption by the operating equipment; and
- Operation schedules of gas- and electricity-fired equipment of industrial enterprises are not constant in time and depend on many factors related both to the intensity of electricity and natural gas consumption, and to the on/off schedules of production and auxiliary equipment [72–74].

In most cases, small, distributed generation systems are fired by natural gas. Moreover, most of the centralized power plants operating in the European part of Russia and the Urals also use natural gas as the main fuel. Figure 8 below shows the diagrams of daily production and generation of electrical and heat energy, as well as relative fuel consumption

of a typical district combined heat and power plant (CHPP). The electricity generation curves are very uneven, seasonally as well as over shorter periods, such as a month or a week. This is due to the specifics of the demand for electricity in the district and integrated energy systems in which the CHPP operates.

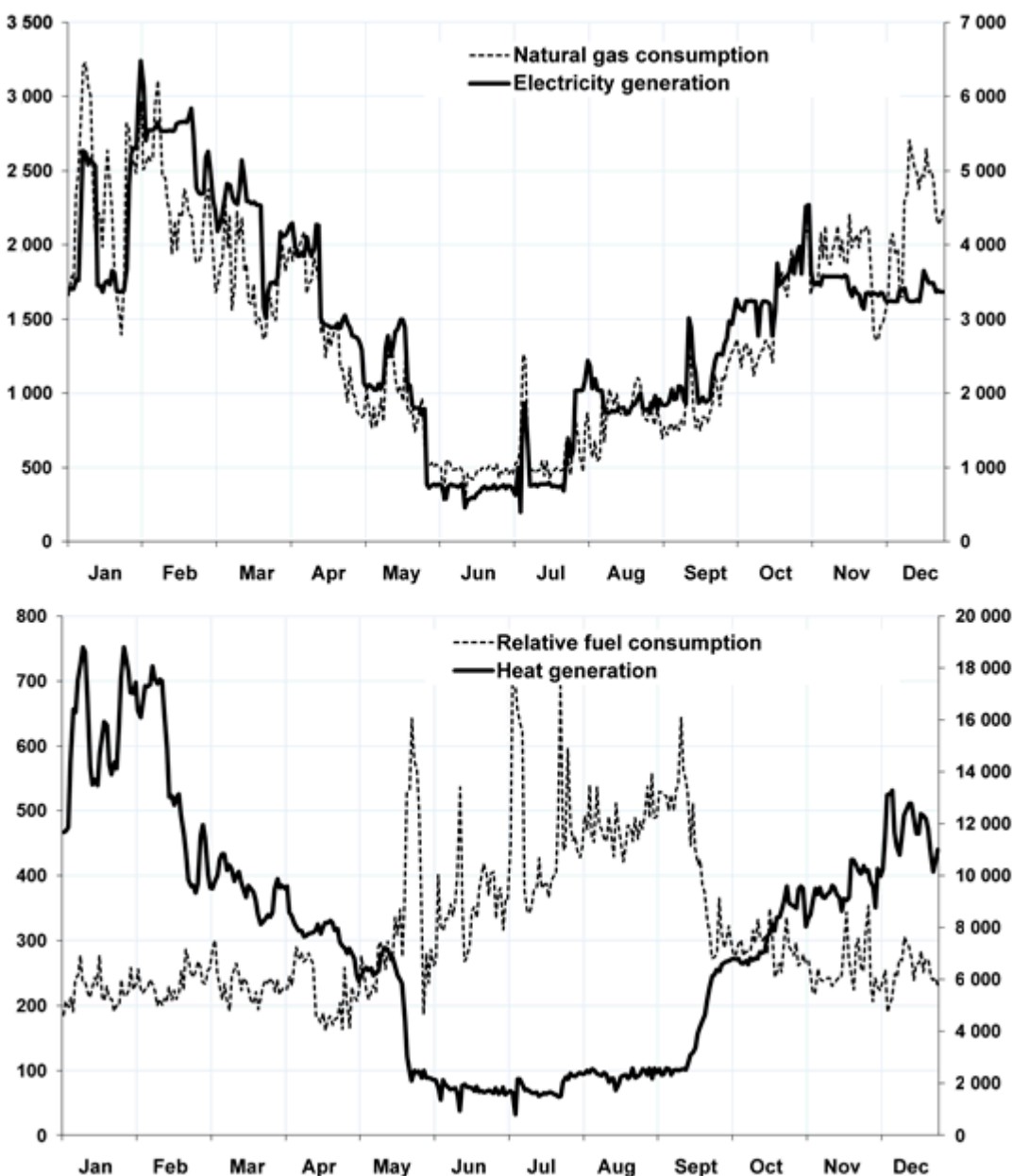

**Figure 8.** Daily electricity and heat generation curves of a typical district CHPP.

The CHPP's gas demand curve is consistent with the operation of its gas-fired equipment such as steam boilers, which drive generators to produce electricity while generating thermal energy for district heating. Synchronous change of the heat and power generation curves leads to changes in the gas consumption from the centralized gas network.

District CHPPs have their annual heat generation curves go hand in hand with the seasonal demand, as its peak falls after the colder seasons of the year. This factor should be taken into account when managing DER systems that produce not only electrical but also thermal energy.

In addition, as follows from Figure 8, the relative fuel consumption of a typical district CHPP increases with a decrease in the plant's capacity utilization and decline as the capacity

utilization increases. This is due to the impact of the power plants' utilization factor—the closer it is to 1, the lower the relative indicators of fuel consumption.

Figure 9 below shows hourly consumption of natural gas and hourly electricity generation of a small distributed generation system. As follows from the graphs, the hourly curves of natural gas consumption by a distributed generation system are also characterized by unevenness, and their shape is similar to the hourly daily electricity consumption curve. This is explained by the fact that the hourly electricity generation curves of distributed generation systems adjust to the demand of local consumers served by the power plants.

To summarize, the consumption of natural gas by industrial enterprises and DER systems integrated into their energy systems is characterized by uneven demand curves, which should certainly be taken into account when managing the operation of DER systems at industrial facilities.

## 4. The Impact of Natural Gas Demand Volatility on Industrial Electricity Pricing

In Russia, industrial enterprises can purchase natural gas from regional gas suppliers or at the gas mercantile exchange. In most regions, Gazprom Mezhregiongaz (subsidiaries of Gazprom) plays the role of a regional gas supplier. At the gas exchange, companies can purchase the commodity through organized trading in the Natural Gas section of the St. Petersburg International Mercantile Exchange, where natural gas is traded at exchange prices by independent providers such as Lukoil, Tatneft, Novatek, etc.

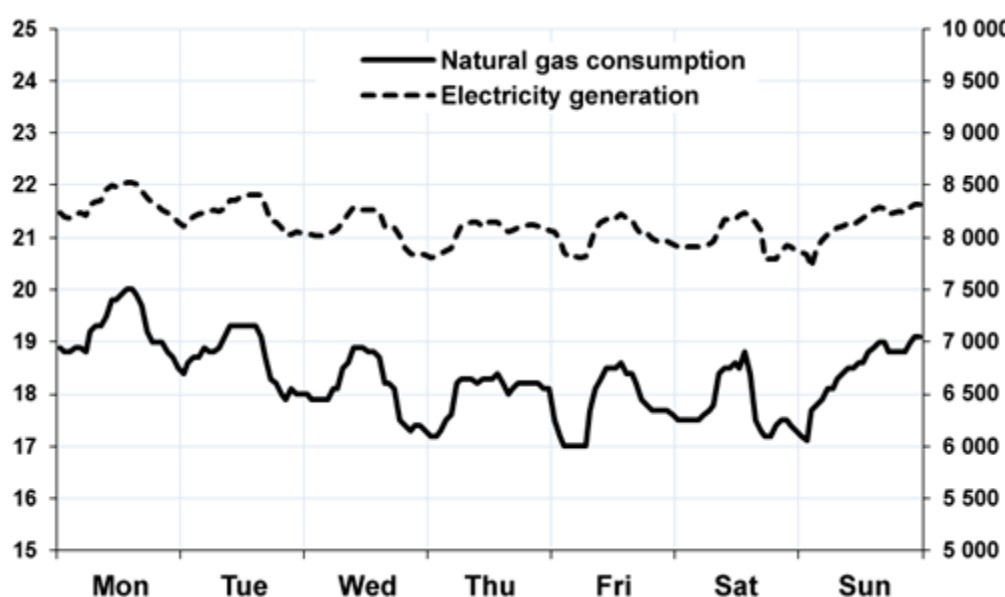

**Figure 9.** Hourly consumption of natural gas and hourly electricity generation of a small distributed generation system.

The purchase price of the gas from a regional supplier consists of several components such as production costs, shipping costs and distribution costs. The key components of the gas price paid by industrial enterprises are the so-called limits, or negotiated thresholds, agreed between suppliers and consumers of natural gas. The purchase price of natural gas for an industrial enterprise is calculated using Formula (1) below.

$$S_{GAS\_m} = S_{GAS\_thrsh\_m} + S_{GAS_{beyond}thrsh\_m} + S_{GAS\_exch\_m} \qquad (1)$$

where:

$S_{GAS\_m}$ is the price natural gas purchased by an industrial enterprise in month m (in USD).

$S_{GAS\_thrsh\_m}$ is the price of natural gas purchased by an industrial enterprise in the regional market within the threshold value in month m (in USD) Formula (2).

$S_{GAS_{beyond}thrsh\_m}$ is the price of natural gas paid by an industrial enterprise on the regional market beyond the threshold value in month m (in USD).

$S_{GAS\_exch\_m}$ is the commodity exchange price of natural gas paid by an industrial enterprise in month m (in USD).

$$S_{GAS\_thrsh\_m} = V_{GAS\_thrsh\_m} \times P_{GAS\_thrsh\_m} \qquad (2)$$

$V_{GAS\_thrsh\_m}$ is the paid volume of natural gas within the negotiated threshold in month m (in '000 cubic meters) Formula (3).

$P_{GAS\_thrsh\_m}$ is the price of gas paid within the negotiated threshold in month m (in USD per '000 cubic meters) Formula (4).

$$V_{GAS\_thrsh\_m} = V_{GAS\_m} \ni \begin{bmatrix} V_{GAS\_m} \leq 1.1 \times V_{GAS\_contr\_m} \\ V_{GAS\_m} \geq 0.8 \times V_{GAS\_contr\_m} \end{bmatrix} \qquad (3)$$

where:

$V_{GAS\_m}$ is the volume of natural gas consumed in month m (in '000 cubic meters).

$V_{GAS\_contr\_m}$ is the volume of natural gas consumption in the contract with the regional supplier in month m (in '000 cubic meters)

$$P_{GAS\_thrsh\_m} = P_{suppl} + P_{ship} + P_{distr} \qquad (4)$$

where:

$P_{suppl}$ is the gas supply rate (in USD per '000 cubic meters).

$P_{ship}$ is the gas shipping rate (in USD per '000 cubic meters).

$P_{distr}$ is the gas distribution rate (in USD per '000 cubic meters).

As follows from Formula (5) below, the volume of gas in the contracted threshold includes the amount consumed in month m, which neither exceeds the negotiated threshold in month m by 10% nor is below it by 20%. If the consumption of natural gas by a given industrial enterprise falls out of this range, the consumer is said to go beyond the threshold.

$$S_{GAS_{beyond}thrsh\_m} = [V_{GAS\_m} \not\ni V_{GAS\_thrsh\_m}] \times P_{GAS_{beyond}thrsh\_m} \qquad (5)$$

where:

$P_{GAS_{beyond}thrsh\_m}$ is the price of gas paid beyond the negotiated threshold in month m (in USD per '000 cubic meters) Formula (6).

$$P_{GAS_{beyond}thrsh\_m} = \begin{array}{l} [\text{period from } 15/04 \text{ to } 15/09 \,], \text{ hence } P_{GAS\_thrsh\_m} \times 1.1 \\ [\text{period from } 16/09 \text{ to } 14/04 \,], \text{hence } P_{GAS\_thrsh\_m} \times 1.5 \end{array} \qquad (6)$$

Thus, the volume of gas that is more than 110% of the agreed volume will be paid for by the industrial enterprise at a 10% higher rate for the period from April 15 to September 15 and at a 50% higher rate for the period from September 16 to April 14. If the enterprise consumes less than 80% of the negotiated volume (despite the fact that natural gas was not in reality consumed), it pays a penalty of 10% of the gas price for the period between April 15 and September 15, and 50% of the price of the non-consumed gas for the period between September 16 and April 14. Given the high penalty markup, especially in the fall and winter period between April 15 and September 15, the beyond threshold gas consumption can significantly increase the company's gas costs and the amounts of gas consumed in its DER system.

Another important parameter when estimating the contract-covered (threshold) and beyond threshold volumes is the estimated daily gas demand thresholds, which are calculated using Formula (7) below:

$$P_{GAS\_thrsh\_dd} = P_{GAS\_thrsh\_m} / n_{dd\_mo} \qquad (7)$$

where:

$P_{GAS\_thrsh\_dd}$ is the daily contract-covered gas threshold (in '000 cubic meters), $n_{dd\_mo}$ is the number of days in month m.

Figure 10 shows an example of the calculation of negotiated and beyond-the-threshold volumes of natural gas consumption. The negotiated (contract-covered) daily gas volume is 10 units (according to Formula (7) above, spread equally among all days). The no-markup range for exceeding the threshold is 10% (1 unit in this example). In the period between the 21st and the 31st, the enterprise exceeded the daily thresholds.

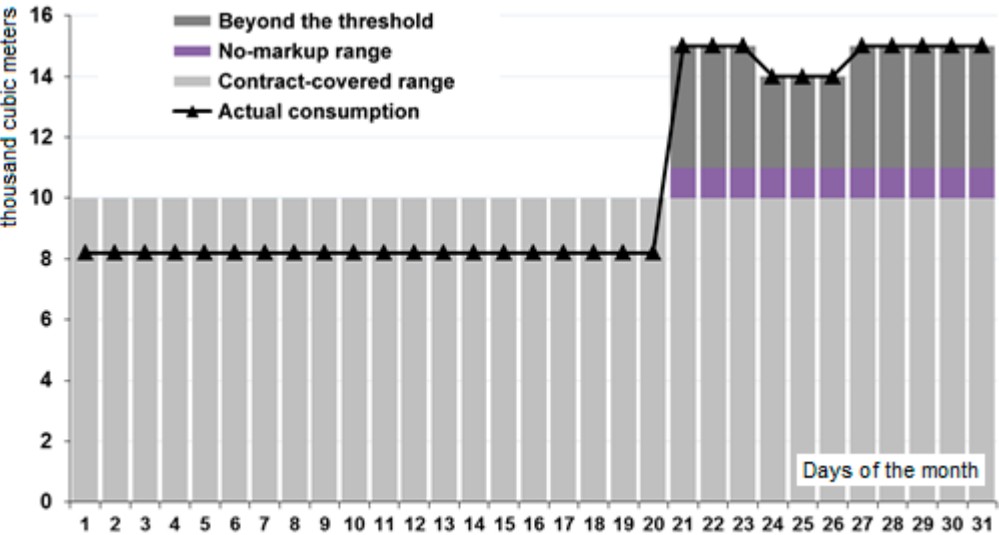

**Figure 10.** Example of the calculation of negotiated and beyond the threshold volumes of natural gas consumption.

Thus, in the period between the 1st and the 20th day, the company's gas consumption was below the negotiated threshold but still within the no-markup range (i.e., not less than 80% of the contract-covered volume), whereas the exceedance of daily thresholds is subject to a penalty.

The volume of natural gas to be purchased at the commodity exchange $S_{GAS\_exch\_m}$ is estimated based on the volume agreed the exchange contract. The exchange contract volumes have priority, i.e., if the company goes beyond the negotiated thresholds $P_{GAS_{beyond}thrsh\_m}$, it can back it up by purchasing natural gas at the gas exchange. The exchange prices must be less than the prices of either contract-covered volumes or the beyond threshold volumes.

Purchasing natural gas from the commodity exchange makes sense in two cases Formulas (8) and (9):

$$P_{GAS\_exch\_m} < P_{GAS\_thrsh\_m} \tag{8}$$

$$P_{GAS\_exch\_m} < P_{GAS_{beyond}thrsh\_m} \tag{9}$$

where:

$P_{GAS\_exch\_m, h}$ is the price of natural gas traded at the commodity exchange in month m (in USD per '000 cubic meters).

## 5. Review of the Drivers Affecting the Natural Gas Demand Fluctuations in Industry

Similarly to electricity, a company's demand for natural gas is impacted by several drivers, including production and regime-related, i.e., related to the operation schedules and regimes of the company's gas-fired equipment. Meteorological factors also impact the operational schedule of the gas-fired boilers at industrial enterprises. The social and economic factors are associated with the work shifts and the alternation of working days

and days off. These drivers also affect the operation schedules of gas-consuming equipment and the regimes of instant gas demand.

In terms of the drivers impacting gas demand by a DER system, we distinguish external and internal factors. External drivers are related to the gas network and the DER system's infrastructure as well as gas prices, whereas internal factors are associated with the operation of the company's equipment, its production process schedules, capacity and parameters of its DER system, and demand response capability.

Figure 11 shows a diagram of the key drivers behind the demand for natural gas by an industrial DER system. Accounting for all internal and external factors, such as the price, demand parameters, system limitations and capacity of the gas network, gas consumption regimes of the enterprise and generator in particular, is essential in order to effectively manage an industrial DER system and adjust its operations for maximizing value and ensuring the system's stability and reliability of operation.

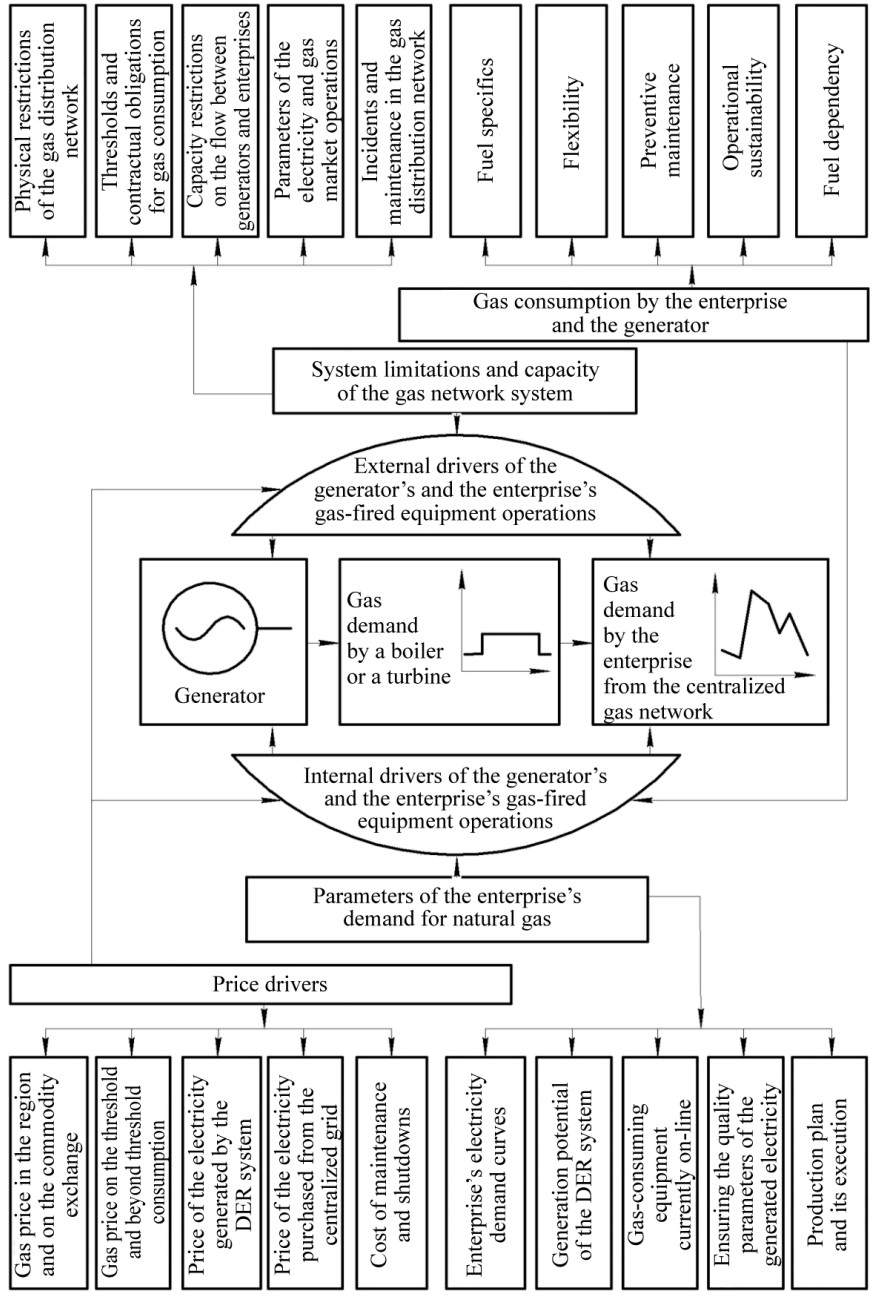

**Figure 11.** Diagram of the key drivers behind the demand for natural gas by a DER system.

## 6. Proprietary Model for the Management of DER Systems in Integration with the Gas Demand Response

Based on the study of the industrial gas pricing, the specifics of distributed generation systems and external and internal drivers associated with the gas market, we developed a model for managing DER systems combined with the gas demand response as outlined in Figure 12 below.

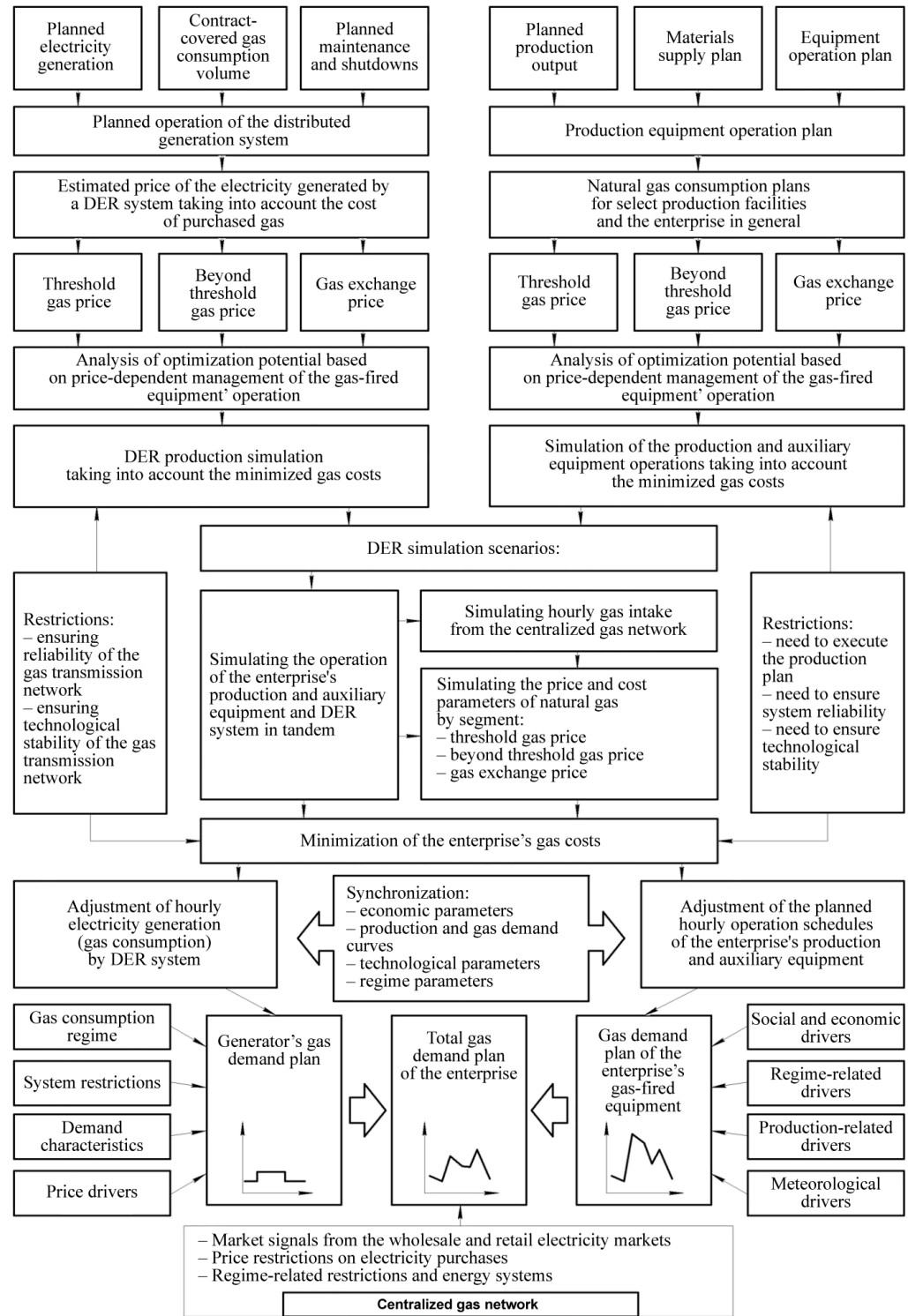

**Figure 12.** Proprietary Model for the Management of DER Systems in Integration with the Gas Demand Response.

The Model analyzes the use of the natural gas in two main areas—internal demand of the enterprise in general and the specific demand of its DER system.

Analysis of the enterprises' internal gas demand includes the analysis of its production plans, operation schedules and materials supply plans. Based on these data, the Model generates hourly demand plan for the enterprise in general and its individual facilities. The hourly demand plan is then used to plan optimal procurements of natural gas and its price from different market segments (threshold volumes, beyond threshold, from the commodity exchange).

Based on the obtained planned cost of natural gas, the Model analyses the possibilities for optimizing the costs of gas purchases based on price-dependent management of gas consumption schedules and modeling of various options for changing the production equipment utilization, with a view to minimizing gas costs.

The Model also analyses gas consumption by the distributed generation system of the enterprise based on the analysis of hourly plans for electricity generation, contractual gas consumption volumes, maintenance and shutdown plans, and generation plans for internal needs. The produced generation plans are then used to forecast the price parameters of natural gas for the DER system by key cost segment. The obtained cost parameters of the electricity generation serve as basis for the analysis of possibilities for optimizing the cost of natural gas consumption and modeling of various schedules of electricity generation with a view to minimizing gas costs of the DER system.

Based on the data from the analysis of the two main areas of modeling, the Model generates scenarios for the consumption of natural gas by the enterprise and its DER system managed in tandem.

As part of the operation of the active energy complex, modeling of equipment operation schedules, price-dependent management of hourly natural gas consumption schedules, modeling of natural gas purchase price parameters for various options for natural gas demand schedules according to the criteria for minimizing natural gas purchase costs are carried out. Modeling is carried out taking into account the existing restrictions in the form of the need to fulfill the production plan of the enterprise, ensure the system reliability of the ente"pris' and the technological stability of the equipment of an industrial enterprise and a distributed generation system.

Based on the results obtained, the planned hourly work schedules of the production and auxiliary equipment of an industrial enterprise are adjusted, the schedules for hourly power generation by the distributed generation system are planned, and the planned parameters of the enterprise and the distributed generation system are synchronized according to economic parameters, power generation consumption schedules, technological parameters and regime parameters. Electricity consumption not covered by the generation of the distributed generation system is consumed from the Unified Energy System under the terms of electricity supply from the wholesale or retail electricity markets, which is taken into account when modeling the operation of an active energy complex.

The proprietary model for the management of DER systems in integration with the gas demand management analyzes the internal and external parameters of the industrial entity's operations and its DER system, as well as factors existing in the integrated distributed energy system where the consumer is able to buy natural gas in various market segments. As part of the functioning of the model, planning of systems for the operation of an industrial enterprise and distributed generation is carried out, forecasting the demand for natural gas consumption, modeling various scenarios for the operation of individual elements of the system and the active energy complex as a whole, associated with a direct or indirect impact on natural gas consumption.

Given the complex structure of the functioning of the active energy complex, the mechanism for its control should simultaneously take into account five key areas:

(1)  accounting for the operation of energy-consuming equipment of an industrial enterprise and generating equipment of a system of small, distributed generation of electricity within a single control system;

(2) accounting for the simultaneous consumption of electrical energy and natural gas by the system of the active energy complex of an industrial enterprise, during the operation of which, within the framework of a single technological system, there is mutual influence and a change in demand schedules for the consumption of electrical energy and natural gas;

(3) comprehensive accounting of both the consumption of electrical energy and natural gas by the system of an industrial enterprise, and the simultaneous generation of electrical energy by a system of small distributed generation and the consumption of natural gas for the needs of electricity generation within the framework of a unified technological and economic management system;

(4) comprehensive accounting of dynamic hourly price indicators of both the external environment of the wholesale and retail electricity (capacity) markets, regional conditions for natural gas supplies and price indicators of natural gas supply on the commodity exchange, and price indicators of the internal environment of the active energy complex in terms of dynamic hourly price indicators of electricity generation by the active energy complex system into the internal network of an industrial enterprise;

(5) taking into account the limitations of managing an active energy complex, such as: ensuring the conditions for the economic efficiency of the operation of an industrial enterprise, taking into account the conditions for ensuring the technological stability of the operation of the technological equipment of an industrial enterprise and a small distributed generation system, taking into account the system reliability of power supply and the operation of an enterprise operating within a single technological process with external energy supply infrastructure within the framework of the Unified Energy System and the Unified Gas Supply System.

Dynamic price indicators are understood as constantly changing price parameters of the cost of purchases and production of electrical energy and the cost of purchases of natural gas by an industrial enterprise in various directions of possible purchases and generation of energy resources.

Dynamic change in price indicators of purchased and produced energy resources of an industrial enterprise is also possible under the influence of two main directions:

(1) the direction of the external environment—the environment of the wholesale and retail electricity markets and the conditions for the supply of natural gas from a regional supplier and the supply of natural gas from the commodity exchange. The price indicators of these pricing mechanisms are characterized by dynamic changes depending on the hours of the day, days of the week, types of days, seasons, periods of planned peak hours of the energy system, periods of the actual hourly maximum of the regional electric power system, introduction of systemic restrictions by the electric grid and gas infrastructure, etc. An active energy complex cannot influence the change in the dynamic change in price indicators acting from the external environment.

(2) the direction of the internal environment—the environment of schedules of own hourly, daily demand for the consumption of electricity and natural gas of an industrial enterprise, on the basis of which pricing is carried out for the indicators of purchased electricity and natural gas. Taking into account the fact that schedules of demand for energy resources are characterized by volatility, their characteristics influence the dynamic change in price indicators for the purchase of energy resources by an enterprise. Unlike the direction of the external environment, an active energy complex has the ability to manage changes in price indicators acting from the factors of the internal environment.

Thus, the control system of the active energy complex is based on the integrated demand management for the consumption of electricity and natural gas from the Unified Energy System and the Unified Gas Supply System, and the integrated synchronous control of the operating parameters of an industrial enterprise and a small distributed generation

system operating in a single technological mode and a single economic system estimating the complex costs of energy consumption.

## 7. Proprietary Mechanism for Managing industrial DER systems in Integration with Demand Response

We have developed a proprietary mechanism for the management of DER systems in integration with the electricity and gas demand response management. The proprietary tool of DER system management is based on the centralized DER control system, which combines performance analytics, operational scheduling of production and DER system, price planning for the wholesale and retail power markets and regional gas markets and exchange, monitoring all elements of the system, and assessment of different active energy management scenarios under various external and internal conditions impacting production and energy demand.

The Tool assesses the dynamic parameters of the full cycle of electricity and natural gas circulation in a given industrial enterprise, including starting from the consumption of electricity and natural gas by the enterprise's equipment and the generation of electric and thermal energy by its DER system for internal auxiliary use as well as for the external grid supplying other electricity consumers. The separate analysis of the consumption of electricity and natural gas by a given industrial DER takes into account the mutual influence of changes in the demand for electricity and natural gas, which are managed in tandem.

The generated electricity and gas demand plans are then used to review the company's production plans, budget and economic indicators of its production as well as the effect of various drivers (social, economic, regime-related, production-related, meteorological), and to identify the equipment which has the most effect on the demand plans of the given industrial enterprise.

The Tool also analyzes DER generation plans and estimates the cost of the electricity generated by the DER system under different operation scenarios as well as assesses the cost drivers impacting the gas demand.

It should be emphasized that the core advantage of a DER system with demand response is its capacity to substitute for a centralized grid; therefore, given the volatility of the electricity demand, all electricity procurement scenarios are made taking into account equipment utilization scenarios as well as DER generation scenarios. This allows more flexibility in managing the company's electricity demand from the centralized grid and results in the associated cost reduction.

The analysis of equipment utilization scenarios and integration of electricity and gas scenarios with the DER generation scenarios includes the analysis of electricity and gas procurement given the existing terms and options. Analysis of the electricity procurements involves an in-depth analysis of each component (electricity price, capacity price, transmission fee), taking into account the specific local rates and parameters which impact the value of each component. A similar analysis is carried out for the gas procurement scenarios taking into account the threshold and beyond threshold volumes as well as volumes to be purchased on the commodity exchange. It should be noted that since distributed generation systems operate mainly on natural gas, the costs associated with natural gas consumption must always be taken into account in generation scenarios. Simulation of equipment operation scenarios must also take into account the limitations of the given DER system, i.e., the need to provide operational sustainability of production equipment and DER system as well as between the DER system and the external grid, and the economic value derived from cost saving as well as from maintaining other cost parameters (operating costs, logistic costs, storage costs, labor costs, including night shifts).

When choosing the optimal scenario for the operation of the DER system with demand response given the existing limitation and capacity, the Tool simulates changes to the production and auxiliary equipment utilization schedules and the DER system's operation schedule and regime.

It takes from one day up to several months to generate the above scenarios. The base period in the scenarios is one day; however, plans are generated on an hourly basis (this is due to the fact that the calculation of price parameters for the purchase of electricity in the wholesale and retail power markets is carried out with hourly discreteness). Each day, the enterprise's integrated management system controls not only its operational parameters but also economic indicators of the DER system, taking into account any adjustments to the production plan, any maintenance and shutdowns, electricity price estimation and monitoring, with the adjustments made as necessary.

## 8. Overview of the Testing Case Study

We tested our model in a typical machine-building facility operating in Perm Region, Russia. The company is engaged in mechanical processing and assembly of products for the oil and gas industry. The enterprise consumes electrical energy for industrial and domestic needs, and also consumes natural gas for the needs of the production equipment. The company purchased and installed a 4000 kW gas-fired distributed generation system, but did not use the system prior to our testing. The enterprise uses natural gas to power its production.

Tables 1–3 below show the outcomes of testing our model of industrial DER system management in integration with the electricity and gas demand response management in the context of an industrial facility. The testing validated the efficiency of our developments.

**Table 1.** Testing results of the Proprietary Model for the Management of DER Systems in Integration with the Electricity Demand Response.

| No. | Parameter | Unit of Measurement | Before DER System | After DER System | Difference |
|---|---|---|---|---|---|
| 1 | Total electricity consumption by the enterprise | MWh | 7000 | 7000 | 0 |
| 2 | Total electricity consumption from the centralized grid | MWh | 7000 | 4500 | −2500 |
| 3 | Total electricity consumption from the DER system | MWh | 0 | 2500 | 2500 |
| 4 | Total electricity generation by the DER system | MWh | 0 | 2500 | 2500 |
| 5 | Total capacity utilization from the centralized grid | MW | 9 | 6 | −3 |
| 6 | Total electricity transmission costs | MW | 10 | 7 | −3 |
| 7 | Price of the electricity component purchased from the centralized grid | USD/ MWh | 1900 | 1900 | 0 |
| 8 | Price of the capacity component purchased from the centralized grid | USD/ MW | 850,000 | 850,000 | 0 |
| 9 | Transmission services | USD/ MW | 1,050,000 | 1,050,000 | 0 |
| 10 | Price of the electricity component generated by the DER system | USD/ MWh | 1350 | 1350 | 0 |
| 11 | Price of the capacity component generated by the DER system | USD/ MWh | 530,000 | 530,000 | 0 |
| 12 | Cost of electricity distribution from the DER system | USD/ MWh | 0 | 0 | 0 |
| 13 | Total cost of the electricity component purchased by the enterprise | USD | 13,300,000 | 11,925,000 | −1,375,000 |
| 14 | Total cost of the capacity component purchased by the enterprise | USD | 7,650,000 | 6,690,000 | −960,000 |
| 15 | Total outgoing distribution costs | USD | 10,500,000 | 7,350,000 | −3,150,000 |
| 16 | Total cost of electricity purchased by the enterprise | USD | 31,450,000 | 25,965,000 | −5,485,000 |
| 17 | Total price of electricity purchased by the enterprise | USD/ kWh | 4.492 | 3.709 | −0.783 |
| 18 | Total price of electricity purchased by the enterprise | % | 100% | 83% | −17% |

**Table 2.** Testing results of the Proprietary Model for the Management of DER Systems in Integration with the Natural Gas Demand Response.

| No. | Parameter | Unit of Measurement | Before DER System | After DER System | Difference |
|---|---|---|---|---|---|
| 1 | Total natural gas consumption by the enterprise | '000 m$^3$ | 3000 | 2700 | −300 |
| 2 | Total natural gas consumption by the DER system | '000 m$^3$ | 5500 | 4900 | −600 |
| 3 | Total natural gas consumption by the enterprise and its DER system | '000 m$^3$ | 8500 | 7600 | −900 |
| 4 | Daily gas limit established by the centralized gas network operator | '000 m$^3$ per day | 230 | 230 | 0 |
| 5 | Natural gas consumption less the allowed threshold range | '000 m$^3$ | 0 | 0 | 0 |
| 6 | Natural gas consumption beyond the allowed threshold range | '000 m$^3$ | 53 | 0 | −53 |
| 7 | Gas purchased from the commodity exchange | '000 m$^3$ | 0 | 1000 | 1000 |
| 8 | Price of gas from the centralized gas network | USD/ '000 m$^3$ | 5000 | 5000 | 0 |
| 9 | Penalty for underconsumption in the centralized gas network | USD/ '000 m$^3$ | 2500 | 2500 | 0 |
| 10 | Penalty for overconsumption in the centralized gas network | USD/ '000 m$^3$ | 2500 | 2500 | 0 |
| 11 | Price of the gas purchased from the commodity exchange | USD/ '000 m$^3$ | 4500 | 4500 | 0 |
| 12 | Total gas purchased from the centralized gas network within contract-covered thresholds | USD | 42,500 | 30,400 | −12,100 |
| 13 | Total penalties for underconsumption of the contract-covered thresholds | USD | 0 | 0 | 0 |
| 14 | Total gas purchased from the centralized gas network in addition to the contract-covered thresholds | USD | 400 | 0 | −400 |
| 15 | Total gas purchased from the commodity exchange | USD | 0 | 4500 | 4500 |
| 16 | Total gas costs | USD | 42,900 | 34,900 | −8000 |
| 17 | Average gas price | USD/'000 m$^3$ | 5047 | 4592 | −455 |
| 18 | Average gas price | % | 100% | 91% | −9% |

**Table 3.** Total economic effect of the Proprietary Model for the Management of DER Systems in Integration with the Natural Gas and Electricity Demand Response.

| No. | Parameter | Unit of Measurement | Before DER System | After DER System | Difference |
|---|---|---|---|---|---|
| 1 | Total costs for the purchase of energy resources | thousand USD | 74,350 | 60,865 | −13,485 |
| 2 | including electricity | thousand USD | 31,450 | 25,965 | −5485 |
| 3 | including natural gas | thousand USD | 42,900 | 34,900 | −8000 |
| 4 | Reduction of energy resources costs | % | 100% | 82% | −18% |
| 5 | including electricity | % | 100% | 83% | −17% |
| 6 | including natural gas | % | 100% | 81% | −19% |

As follows from the calculations above, the total effect of implementing a DER system with demand response management resulted in the reduction of electricity consumption was 17%, or 5,485,000 rubles per month (USD 90,000), which translates to 0.78 rubles/kWh. Adoption of the DER system also reduced the gas costs by 19%, or 8,000,000 rubles per month (USD 130,000). The overall effect from the application of the DER system integrated

with the electricity and gas demand response management amounted to 18% reduction of the total energy costs, or more than 13,485,000 million rubles per year (more than USD 210,000 million per year). To summarize, for an industrial enterprise with an electrical power consumption of 14 MW, the installation of a 4 MW small distributed generation system with demand response management allows reducing total energy consumption by 19%, saving more than 160 million rubles per year (more than USD 2.6 million annually), as well as recouping the investment costs for the acquisition and installation of the DER system and the associated infrastructure and automation.

The testing has validated a clear economic effect of applying our model on DER systems with demand response in Russia as well as other countries of the world.

## 9. Conclusions

Our concluding considerations on the case study are the following:

Despite the current decarbonization trend in the global energy industry with relatively less expensive and environmentally friendly renewable energy sources, many countries, such as Russia, where the price of the natural gas is relatively low, do not have a solid economic basis for a large-scale energy transition. Therefore, the energy industry in the countries with low cost of electricity generation will follow a different vector in the next 20 to 30 years.

Despite the relatively low cost of electricity generation, Russia experiences a significant shortage of the distribution grid capacity. The country is witnessing an increase in electricity consumption across industrial enterprises, which the existing distribution infrastructure cannot cover. In this situation, the adoption of gas-fired small distributed generation systems presents itself as the most efficient solution as it allows covering the electricity demand while providing additional technological and economic advantages.

Modern small distributed energy resource systems integrated with the demand response management and synchronizing production and energy consumption offer substantial saving potential and present themselves as one of the key areas of modern energy development.

Our analysis of the existing global and Russian research in the field of application of distributed energy systems at industrial enterprises suggest the absence of a methodological approach to the integrated management of the electricity and natural gas costs in the process of managing industrial DER systems.

Our proprietary model for the management of DER systems in integration with the gas demand response management is distinguished by synchronous gas demand management with the enterprise's production equipment and a distributed generation system. The model takes into account the composition and structure of the drivers impacting the consumption of natural gas by an industrial DER system, as well as the specifics of the natural gas pricing in regional gas markets and the commodity exchange.

Our proprietary mechanism for managing industrial DER systems in integration with the electricity and gas demand response management is distinguished by (1) synchronous management of electricity consumption from the centralized grid and from the DER system, when the latter can generate electricity for the enterprise's local network as well as for the centralized grid; (2) synchronous management of gas consumption by the enterprise's equipment and DER system; and (3) synchronization of the cost and physical parameters of the electric energy and natural gas utilized by the enterprise's DER system.

The practical importance of our developments has been confirmed by the case study. The overall effect from the application of a DER system integrated with the electricity and gas demand response management amounted to 18% of the total energy costs, or more than 160 million rubles per year (more than USD 2.6 million per year). The potential saving effect can recoup any investments in the DER system, including construction and installation of the local grid and automation infrastructure, and can be obtained in any country in the world.

**Author Contributions:** All authors have made an equal contribution to this work. Conceptualization, A.D., I.S. and A.S.; methodology, A.D. and I.S.; research, A.D. and A.S.; writing—preparation of the original project, A.D. and I.S.; writing —reviewing and editing, A.D.; visualization, A.D.; author supervision, I.S. All authors have read and agreed to the published version of the manuscript..

**Funding:** We gratefully acknowledge the funding of research by the Ministry of Science and Higher Education of the Russian Federation (the South Ural State University Development Program under the Priority 2030 program).

**Institutional Review Board Statement:** Not applicable.

**Informed Consent Statement:** Approval for the study was not required in accordance with national legislation.

**Conflicts of Interest:** The authors declare no conflict of interest.

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
