# Peer review of "Raising the Resilience of Industrial Manufacturers through Implementing Natural Gas-Fired Distributed Energy Resource Systems with Demand Response"

_sustainability, doi:10.3390/su15108241_

Round 1
Reviewer 1 Report
This article is devoted to a rather relevant topic of studying the directions of implementation of distributed energy systems depending on a combination of factors. The authors demonstrate the specific practical results of this study.
However the reviewer has several comments:
1) Is it advisable to use “resilience” or “sustainability” in the title of the work?
2) The topic of the special issue is “Business Model Innovation for Corporate Sustainability". The authors need to justify in the text of the article (Introduction section) how the topic of your work is related to the topic of the journal
3) Abstract. It is recommended to avoid using abbreviations in this part of the work
4) Introduction, 2 paragraph. The description in the text and the name of Figure 1 do not match
5) Introduction. It is recommended to add a summary of your work
6) Section 8. To understand the specifics of the study, it is necessary to add a description of the company under study, including the specifics of its activities
7) Conclusions. It is recommended to present this section in the form of text, not a bulleted list
8) A Discussion section should be added, in which it is recommended to compare the results obtained by the authors with the results of similar works or to justify why this is impossible
Author Response
This article is devoted to a rather relevant topic of studying the directions of implementation of distributed energy systems depending on a combination of factors. The authors demonstrate the specific practical results of this study.
Authors’ response. Thank you very much for the positive assessment of our article, its deep and professional analysis. Working on your comments helped to significantly improve the article, make it more logical and reasonable. More detailed responses to your comments are given below. All changes in the article are highlighted in yellow.
_____________________________________________________________________________________
1) Is it advisable to use “resilience” or “sustainability” in the title of the work?
Authors’ response: Based on your recommendation, the title of the article has been changed «Improving the economic efficiency of industrial producers through the use of energy distribution systems, the use of natural resources, gas, with consumption».
_____________________________________________________________________________________
2) The topic of the special issue is “Business Model Innovation for Corporate Sustainability". The authors need to justify in the text of the article (Introduction section) how the topic of your work is related to the topic of the journal.
Authors’ response: Corresponding corrections have been made to the text of the introduction of the article.
_____________________________________________________________________________________
3) Abstract. It is recommended to avoid using abbreviations in this part of the work
Authors’ response: Your recommendation is absolutely correct. We have excluded abbreviations in the abstract to the article.
_____________________________________________________________________________________
4) Introduction, 2 paragraph. The description in the text and the name of Figure 1 do not match
Authors’ response: Your recommendation is absolutely correct. The link to Figure 1 has been corrected.
_____________________________________________________________________________________
5) Introduction. It is recommended to add a summary of your work
Authors’ response: We consulted with the editors - this format of introduction is acceptable.
_____________________________________________________________________________________
6) Section 8. To understand the specifics of the study, it is necessary to add a description of the company under study, including the specifics of its activities
Authors’ response: Your recommendation is absolutely correct. A description of the company under investigation has been added to section 8.
_____________________________________________________________________________________
7) Conclusions. It is recommended to present this section in the form of text, not a bulleted list
Authors’ response: Your recommendation is absolutely correct. The conclusions section has been revised and presented in the form of text.
_____________________________________________________________________________________
6) A Discussion section should be added, in which it is recommended to compare the results obtained by the authors with the results of similar works or to justify why this is impossible
Authors’ response: We consulted with the editors - this format of introduction is acceptable.
_____________________________________________________________________________________
Best of Luck!
Once again we thank you for such careful work with our manuscript. We hope we were able to answer your questions.

Reviewer 2 Report
The manuscript is about raising the resilience of industrial manufacturers through implementing natural gas-fired distributed energy resource systems with demand response. The scope of this article is consistent with the requirements of the Sustainability, but it requires major revision in accordance with the comments below:
1. Abstract is too long. According Instruction for Authors should be a total of about 200 words maximum. The abstract should be a single paragraph and should follow the style of structured abstracts, but without headings: 1) Background: Place the question addressed in a broad context and highlight the purpose of the study; 2) Methods: Describe briefly the main methods or treatments applied. Include any relevant preregistration numbers, and species and strains of any animals used. 3) Results: Summarize the article's main findings; and 4) Conclusion: Indicate the main conclusions or interpretations.
2. The introduction is well written, but it might be worth to cite some publications in a similar subject area from Journal Sustainability.
3. Figure 5. It will be clearer if the price of natural gas will be in $ per 1000 cubic metres.
4. Figures 8 and 9. The axes must be described.
5. Section 4, all prices should be converted to $
6. Figure 10. There are missing units on the y-axis.
7. Tables 1 – 3, all prices should be converted to $.
Author Response
The manuscript is about raising the resilience of industrial manufacturers through implementing natural gas-fired distributed energy resource systems with demand response. The scope of this article is consistent with the requirements of the Sustainability, but it requires major revision in accordance with the comments below.
Authors’ response. Thank you very much for the positive assessment of our article, its deep and professional analysis. Working on your comments helped to significantly improve the article, make it more logical and reasonable. More detailed responses to your comments are given below. All changes in the article are highlighted in yellow.
_____________________________________________________________________________________
1) Abstract is too long. According Instruction for Authors should be a total of about 200 words maximum. The abstract should be a single paragraph and should follow the style of structured abstracts, but without headings: 1) Background: Place the question addressed in a broad context and highlight the purpose of the study; 2) Methods: Describe briefly the main methods or treatments applied. Include any relevant preregistration numbers, and species and strains of any animals used. 3) Results: Summarize the article's main findings; and 4) Conclusion: Indicate the main conclusions or interpretations.
Authors’ response: We consulted with the editors - this abstract format is acceptable.
_____________________________________________________________________________________
2) The introduction is well written, but it might be worth to cite some publications in a similar subject area from Journal Sustainability.
Authors’ response: We consulted with the editors - this format of introduction is acceptable.
_____________________________________________________________________________________
3) Figure 5. It will be clearer if the price of natural gas will be in $ per 1000 cubic metres.
Authors’ response: The diagram has been corrected taking into account the comments.
_____________________________________________________________________________________
4) Figures 8 and 9. The axes must be described.
Authors’ response: Figures 8 and 9 are described in section 3 of the text of the article.
_____________________________________________________________________________________
5) Section 4, all prices should be converted to $
Authors’ response: The text has been corrected.
_____________________________________________________________________________________
6) Figure 10. There are missing units on the y-axis.
Authors’ response: Axis units added.
_____________________________________________________________________________________
7) Tables 1 – 3, all prices should be converted to $.
Authors’ response: Units of measure changed
_____________________________________________________________________________________
Best of Luck!
Once again we thank you for such careful work with our manuscript. We hope we were able to answer your questions.

Reviewer 3 Report
1.Please add some quantitative data in the abstract section related to the main research outcomes.
2.China is the largest energy consumer, and I think the author should add Chinese data to fig4 and fig5.
3 The coordinates shall be marked with the scale and the unit in fig7.
4.The novelty of the work must be clearly addressed and discussed, compare your research with existing research findings and highlight novelty.
5.Conclusion section is missing some perspective related to the future research
work, quantify main research findings.
6.What is the impact of wind power, photoelectric and other new energy on the development of natural gas combustion distributed energy system. This article should be discussed,such as https://doi.org/10.1016/j.petsci.2022.09.025.
Author Response
1) Please add some quantitative data in the abstract section related to the main research outcomes.
Authors’ response: We consulted with the editors - this format of the resume section is acceptable.
_____________________________________________________________________________________
2) China is the largest energy consumer, and I think the author should add Chinese data to fig4 and fig5.
Authors’ response: China figures added to figures.
_____________________________________________________________________________________
3) The coordinates shall be marked with the scale and the unit in fig7.
Authors’ response: To the description of Figure 7, the parameters of the axes of the graphs are signed. Applying dimensions directly to the charts will heavily load the charts
_____________________________________________________________________________________
Best of Luck!
Once again we thank you for such careful work with our manuscript. We hope we were able to answer your questions.

Round 2
Reviewer 2 Report
The manuscript can be accept in the present form.